# The visual cortex produces gamma band echo in response to broadband visual flicker

**Alexander Zhigalov** *, **Katharina Duecker**, **Ole Jensen**

Centre for Human Brain Health, School of Psychology, University of Birmingham, Birmingham, United Kingdom

* a.zhigalov@bham.ac.uk

**Data Availability Statement:** Data and scripts are available on the OSF website, https://osf.io/fe8x5/.

**Funding:** The work was supported by the following funding: James S. McDonnell Foundation, Understanding Human Cognition Collaborative

## Abstract

The aim of this study is to uncover the network dynamics of the human visual cortex by driving it with a broadband random visual flicker. We here applied a broadband flicker (1–720 Hz) while measuring the MEG and then estimated the temporal response function (TRF) between the visual input and the MEG response. This TRF revealed an early response in the 40–60 Hz gamma range as well as in the 8–12 Hz alpha band. While the gamma band response is novel, the latter has been termed the alpha band perceptual echo. The gamma echo preceded the alpha perceptual echo. The dominant frequency of the gamma echo was subject-specific thereby reflecting the individual dynamical properties of the early visual cortex. To understand the neuronal mechanisms generating the gamma echo, we implemented a pyramidal-interneuron gamma (PING) model that produces gamma oscillations in the presence of constant input currents. Applying a broadband input current mimicking the visual stimulation allowed us to estimate TRF between the input current and the population response (akin to the local field potentials). The TRF revealed a gamma echo that was similar to the one we observed in the MEG data. Our results suggest that the visual gamma echo can be explained by the dynamics of the PING model even in the absence of sustained gamma oscillations.

## Author summary

The properties of the neuronal dynamics governing the visual system are highly debated. While some emphasize the neuronal firing rate and evoked activity in response to visual stimuli, others emphasize the oscillatory neuronal dynamics. To investigate the dynamical properties of the visual system, we recorded the magnetoencephalography while stimulating the visual system using a broadband (1–720 Hz) visual flicker. By estimating the temporal response function (similar to cross-correlation) between the visual input and neuronal activity, we demonstrated a clear response in the gamma band that we term the gamma echo. We then constructed a physiologically realistic network model that could generate gamma-band oscillations by a pyramidal-interneuron gamma (PING) mechanism. This model allowed us to account for empirically observed response in the gamma band, and to provide novel insight on the neuronal dynamics governing the early visual

Award, grant number 220020448, www.jsmf.org, to O.J.; the Wellcome Trust Investigator Award in Science, grant number 207550, wellcome.org, to O.J.; Biotechnology and Biological Sciences Research Council, grant BB/R018723/1, bbsrc.ukri.org, to O.J.; Wolfson Research Merit Award, Royal Society, royalsociety.org, to O.J. The funders had no role in study design, data collection and analysis, decision to publish, or preparation of the manuscript.

**Competing interests:** The authors have declared that no competing interests exist.

system. The stage is now set for further investigating how the gamma echo is modulated by tasks such as spatial attention as well as uncovering how the echo might propagate in the visual hierarchy.

## Introduction

The properties of the neuronal dynamics governing the visual system are highly debated. Some emphasize the neuronal firing rate [1–3] and evoked activity [4] in response to visual stimuli. Others emphasize the oscillatory neuronal dynamics. In particular, neuronal oscillations in the gamma band have been proposed to bind visual features by means of synchronized spiking [5,6] as well as supporting communication between different brain regions [7,8].

In this paper we are applying a new tool for investigating the dynamical properties of the visual cortex in humans. We are making use of a new type of LED/DLP projector (Propixx, VPixx Technologies Inc., Canada) that has a refresh-rate of up to 1440 Hz. The projector makes it possible to stimulate the visual system with broadband flickering stimuli while measuring the brain response using magnetoencephalography (MEG). This approach allows for estimating the temporal response function (TRF). The TRF is the kernel that best explains the brain response when convolved to the broadband input signal. In other words, the TRF can be considered a simple model capturing the filter properties of the visual cortex. In previous studies, such an approach has been used to investigate the dynamical properties of the visual system at lower frequencies. Using a broadband flicker (1–80 Hz), the TRF was approximated from the cross-correlation between the EEG and the input signal [9]. This approach revealed a robust response in the alpha range termed "the perceptual echo". Yet, the authors did not report dynamical properties in the gamma range most likely due to the limited refresh rate of the monitor [9]. The aim of this study was to ask if the TRF also has a band-limited response at higher frequencies, to uncover the faster dynamical properties of the visual system. As we will show, the TRF function has a clear response at higher frequencies which is limited to the gamma band.

The oscillatory dynamical properties of the cortical tissue have also been investigated by means of computational modelling. This has resulted in the notion that neuronal gamma oscillations are generated by the so-called pyramidal interneuron gamma (PING) mechanism [10–12]. According to this mechanism, GABAergic interneurons play an essential role in determining the frequency and synchronization properties for the generation of gamma oscillations. Basically, the decay of the GABAergic feedback is a key variable determining the period of each gamma cycle as the GABAergic hyperpolarization prevents neuronal firing of both pyramidal and interneurons of about 10–20 ms [13]. Furthermore, the GABAergic feedback also serves to synchronize the population activity [14,15]. In each cycle, the firing of the pyramidal cells serve to excite the interneurons thus initiating the next oscillatory cycle. This mechanism was first uncovered in hippocampal rat slices [16] and supported by computational modelling [10–12]. Later, the GABAergic based mechanism was also investigated using optogenetic studies in the somatosensory cortex [17,18] and the visual system [19] in mice. A human MEG study demonstrated that gamma oscillations are strongly modulated after the GABAergic feedback was manipulated by the GABAergic agonist lorazepam in a double-blind study. As predicted by the PING model, the visual gamma oscillations decreased in frequency while they increased in power as the GABAergic feedback increased with the administration of lorazepam [20]. Other studies have reported a link between gamma frequency and the GABA concentration as measured by magnetic resonance spectroscopy (MRS) in both visual and

somatosensory regions [21–23] (but see [24]). Finally a PET study measuring GABA(A) receptor density found a link to gamma frequency [25].

The PING mechanism can be implemented using different biophysical models [26–28]. Several of these implementations are based on Hodgkin–Huxley type of models. In this work, we based our simulations on the Izhikevich model [29] which has reasonable realistic dynamics and computational efficiency of integrate-and-fire neurons. This model is capable to produce variety of spiking dynamics such as regular spiking, fast spiking, low threshold spiking and other by adjusting only four parameters. In contrast to other more complex models, e.g., [30] that require tuning of multiple parameters, the Izhikevich model is relatively simple yet capable of explaining a wide range of phenomena pertaining neuronal synchronization and oscillations [31–33].

In this study, we asked if the dynamical properties of the PING model can account for the gamma response in the TRF we observed in the MEG data. The basic idea was to simulate a network model for gamma oscillations with a broadband signal. This allowed us to estimate the TRF of the network model and relate it to the TRF from MEG study. If the network model can procure a TRF similar to the one observed in the MEG data, then we have provided novel insight on the neuronal dynamics governing the early visual system.

In short, to investigate the dynamical properties of the visual system, we recorded the MEG while stimulating the visual system using a broadband (1–720 Hz) visual flicker. This allowed us to estimate the TRF of the visual system. As we will demonstrate, this resulted in a clear band-limited response in the gamma band. We then constructed a physiologically realistic network model that could generate gamma-band oscillations by a PING-type mechanism. This model allowed us to account for empirically observed TRF in the gamma band.

## Results

We used a moving grating paradigm in which the left and right visual stimuli were generated using orthogonal broadband random signals while we recorded the ongoing MEG.

### Broadband visual stimulation reveals alpha and gamma echoes in the visual system

The TRF of a system can be estimated by deconvolving its input and output. We used ridge regression (see, Materials and Methods) to compute the TRF for the MEG signals from sensors over visual cortex while stimulating with a broadband random visual input. Fig 1 shows the TRF for an occipital sensor for a representative participant. The TFR has a rich temporal structure (Fig 1A, black line). Applying a bandpass filter in the gamma band (40–100 Hz) to the TRF revealed an early response (Fig 1A, blue line) at about 40 ms. A bandpass filter in the alpha band (8–13 Hz) revealed a later response comprising several cycles. A time-frequency analysis of power further demonstrated the presence of band-limited responses in the alpha and gamma band in the TRF (Fig 1B). While the late response–the"alpha perceptual echo"– was reported in the previous studies (e.g. [9], the early response–the"gamma echo"–is so far unobserved property of the visual system.

### TRF show individual resonance frequency of gamma echo

To further evaluate the characteristics of the gamma echo, we computed TRF for five individual participants using the MEG gradiometers with strongest response to the broadband input signal. The frequencies of the individual gamma echoes ranged from 46 to 56 Hz and were

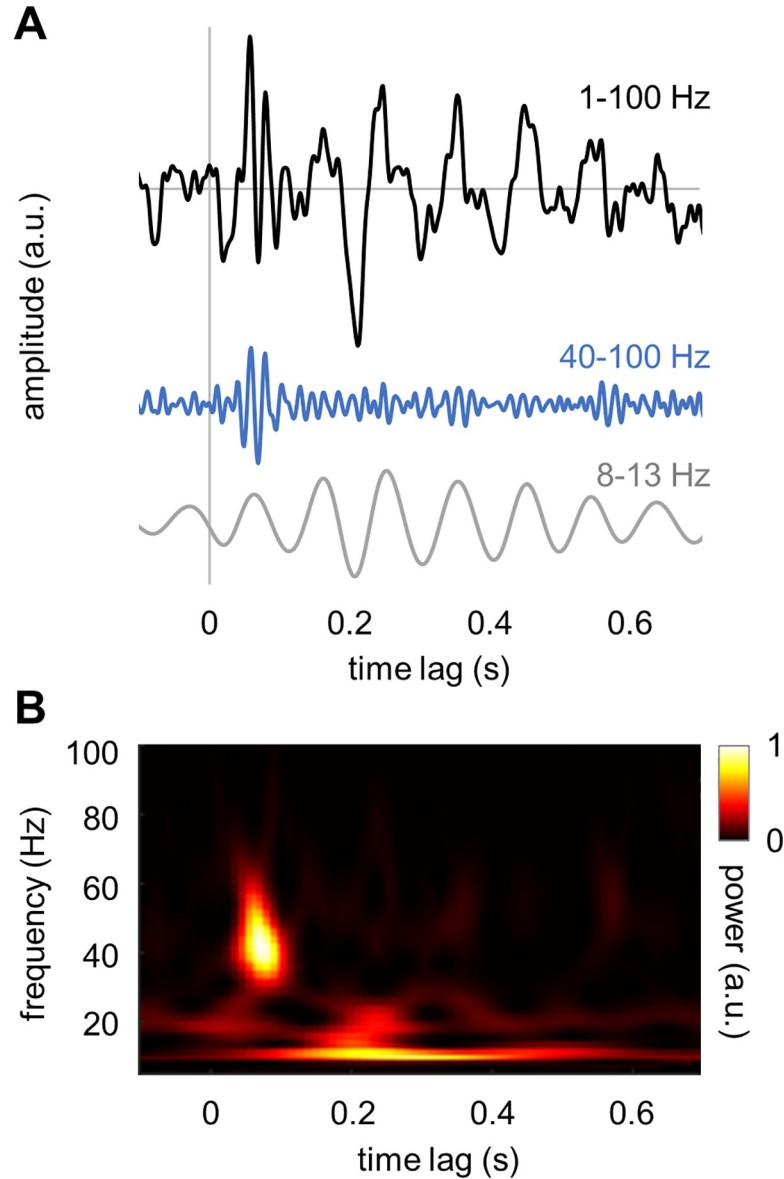

**Fig 1. TRF of the human visual cortex derived from a random broadband visual input train.** (A) The TFR filtered in the broad band (1–100 Hz; black line), the gamma band (40–100 Hz; blue line) and the alpha band (8–13 Hz; gray line). (B) Time-frequency representation of power of the TRF. Note that the TRF was computed for an occipital gradiometer that captures both the alpha band and gamma band TRF.

close to 48 Hz on average (Fig 2). The responses in the gamma band were strongest at 50–100 ms and spanned over 2–3 cycles. To ensure that the 50 Hz line noise does not contribute to the gamma echo, we assessed the power spectral density and TRF for magnetometers before and after applying the source space separation (SSS) method [34] as implemented in MNE Python toolbox [35]. SSS removes artifacts caused by external disturbances such as line noise, and hence, provides possibility to evaluate contribution of 50 Hz noise to the gamma echo. We performed the analysis on the magnetometers as they are particularly sensitive to 50 Hz line noise and hence, they provide a worst-case setting. The results clearly showed that suppression of 50

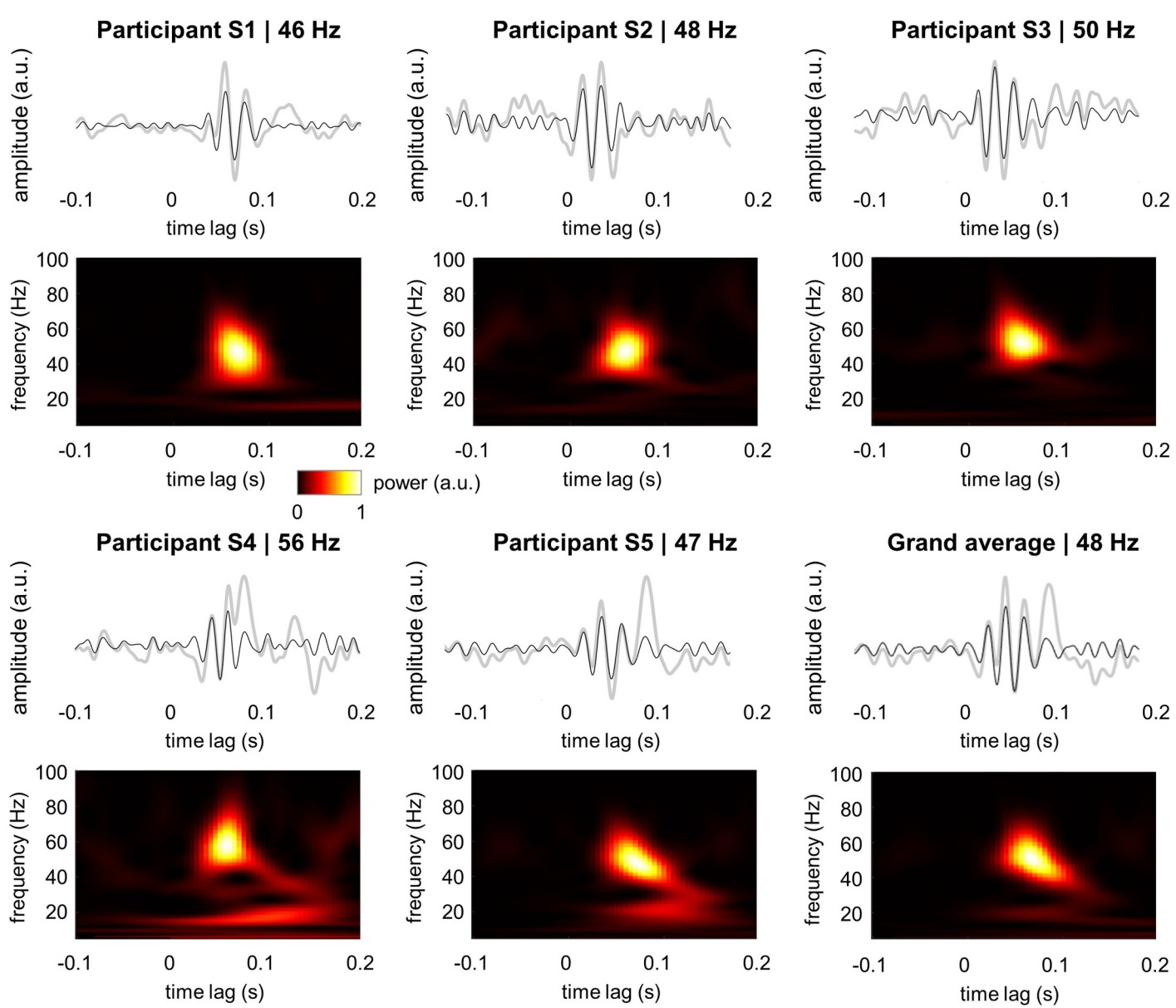

**Fig 2. TRF and the associated time-frequency representation of power for individual participants.** Note the robust response in 40–60 Hz gamma range. Gray lines depict the TFR at 1–100 Hz while the black lines show the response filtered at 40–100 Hz.

Hz noise in data did not change the characteristics of gamma echo (S1 Fig) in any of the participants. This demonstrates that the gamma echo is not biased by the line noise.

## Induced gamma oscillations

To relate the gamma echo response and visually induced gamma oscillations, we computed a time-frequency representation of power at the occipital sensor with the strongest response to flicker (see, Fig 3). The grating induced gamma oscillations at around 57 Hz (grand average). Interestingly, the individual frequencies of the induced gamma oscillations were faster than frequencies of the gamma-echo (Fig 3). These results suggested that the gamma echo and induced gamma oscillations are produced by different generators.

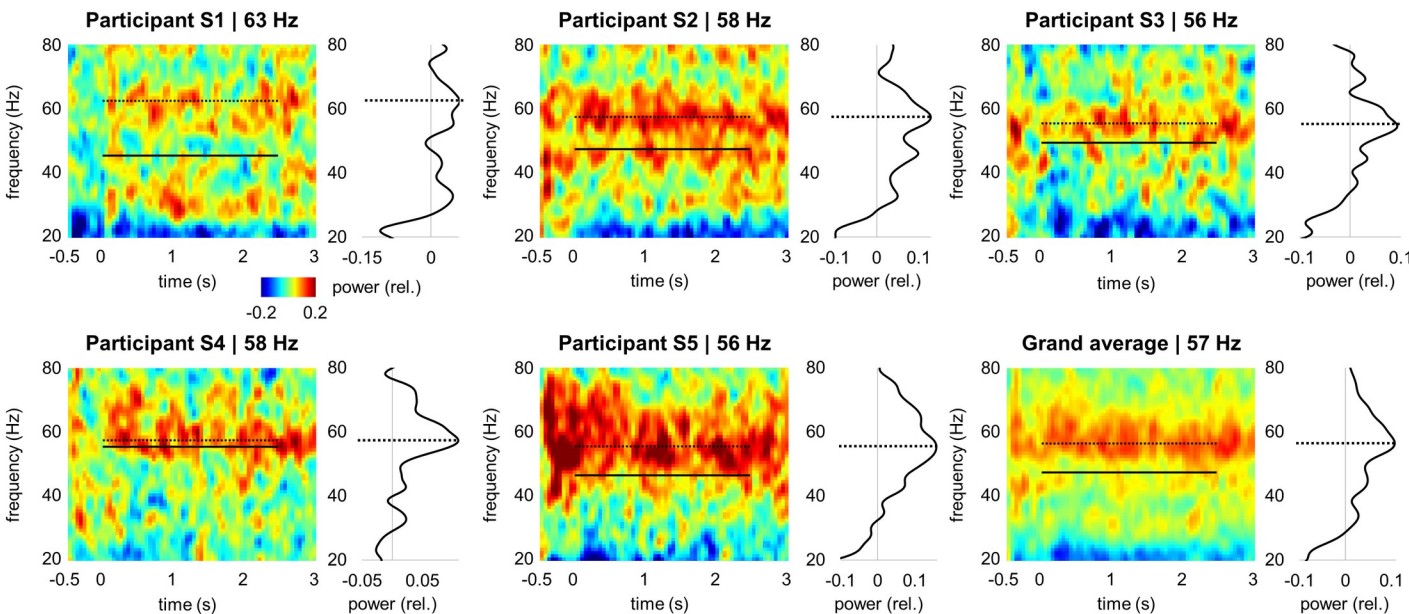

**Fig 3. Time-frequency representation of the power (relative change) induced by the gratings.** The dashed line indicates frequency of the induced gamma oscillations, and for comparison, the solid line indicates frequency of the gamma echo. Curves next to the time-frequency plots represent power averaged over the -0.5 to 3 s time interval.

## Localization of gamma echoes and induced gamma oscillations in sensor and source spaces

To identify the generators of the gamma echo and induced gamma oscillations, we computed their spatial characteristics at the sensor and source levels. Both sensor and source space topographies of the gamma echo clearly showed that the response was mainly localized in the primary visual cortex (Fig 4A and 4B). Similarly, induced gamma oscillations were originated in the visual cortex as suggested by sensor and source space topographies (Fig 4C). The sources of the gamma echo response and induced gamma oscillations were largely overlapped (Fig 4), although, sources of the gamma echo appeared more lateralised and superior. We assessed their spatial overlap by extracting coordinates along z-axis and comparing these coordinates across subjects using t-test (see, Materials and Methods). Although, the sources of induced gamma oscillations were slightly superior (~11 mm on average) compared those of the gamma echo, the difference was not significant ($p > 0.19$, t-test). These results suggest that the gamma echo response and induced gamma oscillations are produced by neighbouring but not necessarily the same sources.

## PING based model of gamma oscillations

We implemented a pyramidal-interneuron gamma (PING) [10,12,16] network model with biologically plausible synaptic currents [11] attempting to account for the TRF in the gamma band. The model consisted of interconnected excitatory (E) and inhibitory (I) cells (Fig 5A). The connectivity matrix in Fig 5B describes the connection strengths between all the cells. In this model, the connectivity strength between different type of cells was set based on prior empirical findings [36], see Materials and Methods. The connection strength was weighted by random values drawn from a uniform distribution [0, 1], to ensure heterogeneous connectivity. For the connectivity matrix in Fig 5B and constant input currents of 12.25 μA and 5.25 μA

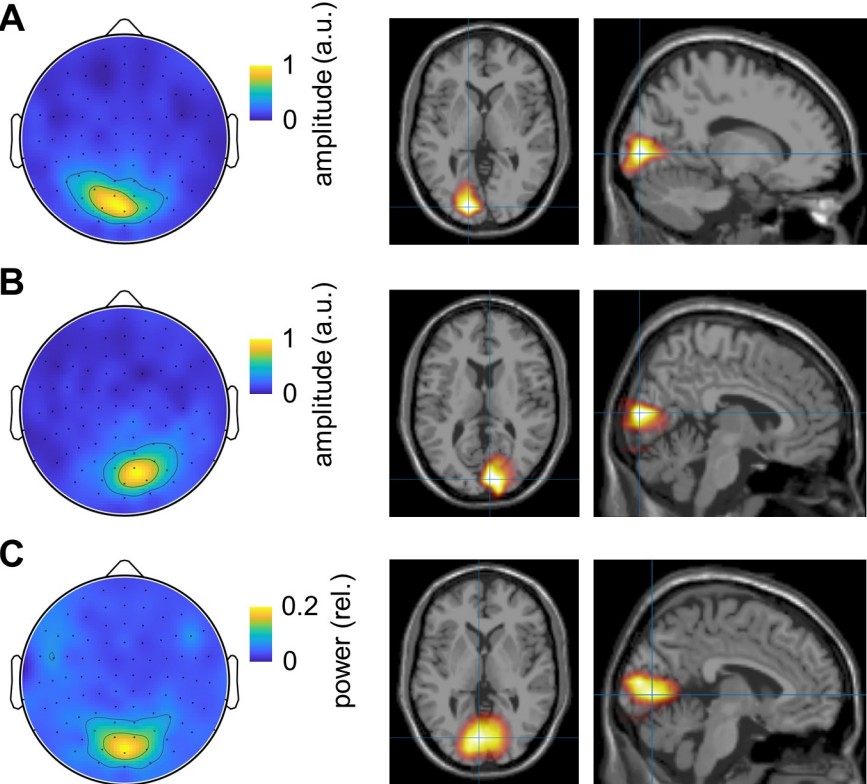

**Fig 4. Topographies and source modelling of the gamma echo and induced gamma oscillations.** (A) The topography and the source modelling (LCMV beamformer) of the peak amplitude of the gamma echo (0.04–0.08 s) for the right flickering stimuli. (B) Same as (A) but for left flickering stimuli. (C) The topography and source modelling of the power (DICS beamformer) of the induced gamma oscillations within range 40–100 Hz.

to the E-cells and I-cells, respectively, the model produced synchronous spiking activity (Fig 5C). The membrane potentials of the excitatory neurons were averaged to approximate the local field potentials, i.e., the population activity (Fig 5D). Spectral analysis revealed robust oscillations in the local field potentials at 48 Hz (Fig 5E). The PSD was computed and subsequently averaged over 20 trials, to reduce random variations in the model output.

In accordance with the PING mechanism, the model firing rate is determined by the input currents and the connectivity strength [11], suggesting that the resonance frequency of 48 Hz can be obtained for different combinations of the input currents and connectivity. To further explore this possibility, we assessed the model output power (at the spectral peak) and corresponding frequency (Fig 6A) by varying the input currents to E-cells and I-cells while preserving the connectivity parameters and the network size. The 2D parameter space diagram (Fig 6B and 6C) indicated that resonance frequency of 48 Hz can be obtained for several combinations of the input currents. Considering this, in addition to the input currents used in our simulations above ($I_E$ = 12.25 µA and $I_I$ = 5.25 µA; black circle in Fig 6B and 6C), we also explored two pairs of the input currents: $I_E$ = 8.75 µA and $I_I$ = 5.25 µA (red circle in Fig 6B and 6C), $I_E$ = 12.75 µA and $I_I$ = 7.25 µA (blue circle in Fig 6B and 6C), which produce oscillations at the resonance frequency. Temporal characteristics of the spiking activity and mean field potentials for three pairs of the input currents are shown in Fig 6D. These results suggested that our initial model parameters ($I_E$ = 12.25 µA and $I_I$ = 5.25 µA) provided more stable oscillations compared to the other pairs of the input currents.

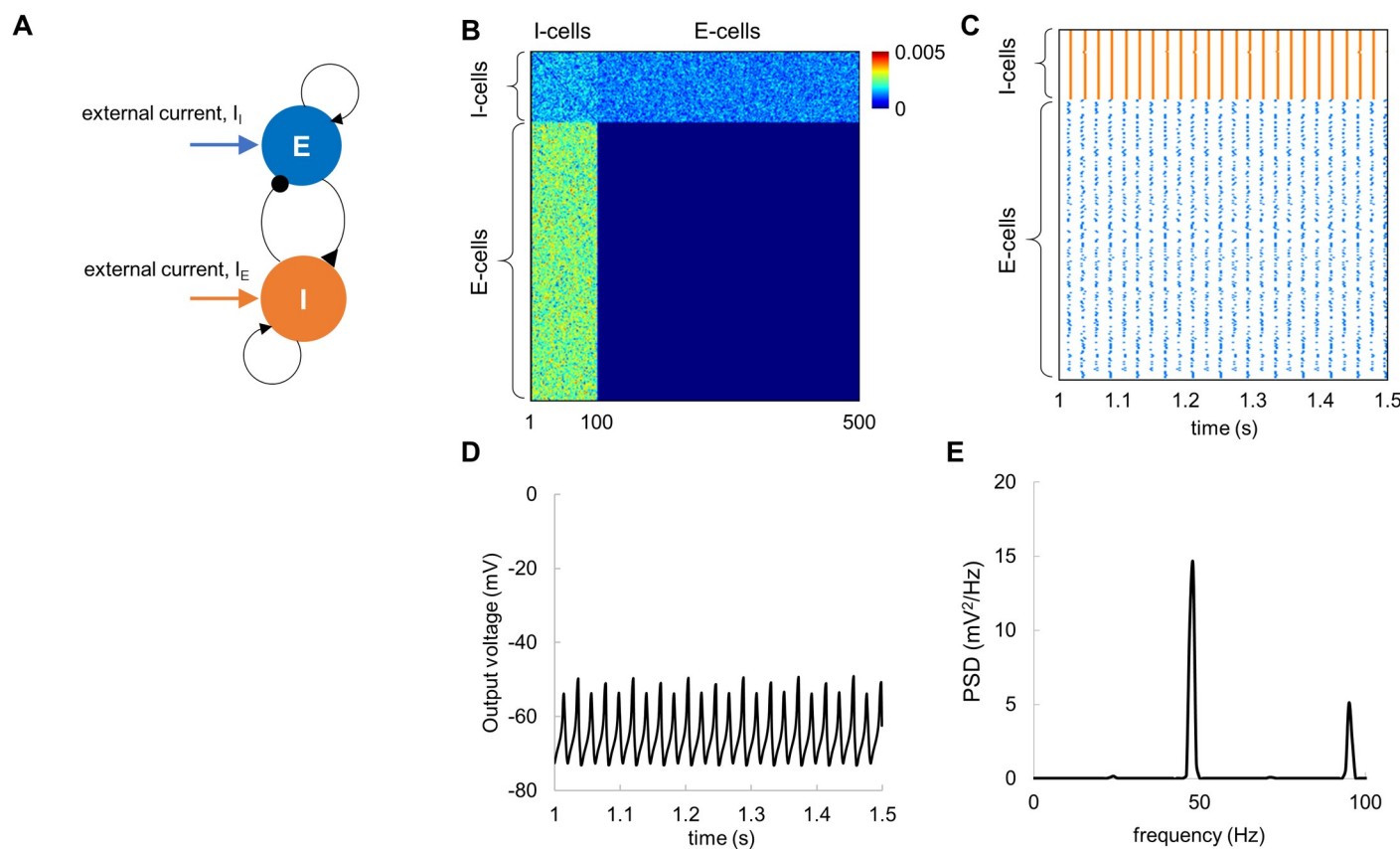

**Fig 5. The PING model with constant input currents produces robust neuronal oscillations at around 48 Hz.** (A) Neuronal architecture; the simulated network consisted of interconnected E-cells (N = 400) and I-cells (N = 100). (B) Connectivity matrix between E-cells and I-cells. (C) Spike rastergram for E-cells (blue) and I-cells (orange) shows temporal synchronization among the cells in the presence of constant input current. (D) The average membrane potential of the E-cells exhibited prominent oscillations. (E) Power spectral density of the average membrane potential for the E-cells shows a clear peak at 48 Hz. Note that PSD was averaged over 20 trials.

To ensure that the model was operating in a stable regime, we estimated the duration of transient effect after the input current was applied (Fig 7A). The time diagram of spiking activity and mean field potentials showed that the transient effect lasted less than 0.5 s. In the further analyses, we discarded the first 1 s of the signal to make sure that the remaining signal is stationary.

Finally, we evaluated the impact of the network size on frequency and power of the mean-field potentials (Fig 7B). Importantly, while changing the network size, we preserved the ratio between E-cells and I-cells as 4 to 1 following our original model [29]. The results suggested that an increase in the network size from 200 to 1000 neurons was associated with a slight decrease in the frequency from 50 to 46 Hz, and an increase in the power. Such a relatively small change in the output characteristics suggested robustness of the model.

## Broadband input to the model produces a gamma echo at the resonance frequency

In order to estimate the TRF of the network dynamics, we applied broadband input current to the model. The input current was modelled as a sum of constant current of 12.25 μA and random (uniform) noise with amplitude of 4 μA to the E-cells and a constant current of 5.25 μA to I-cells (Fig 8A). This simulates the LGN input to the area V1 in the visual cortex. For the

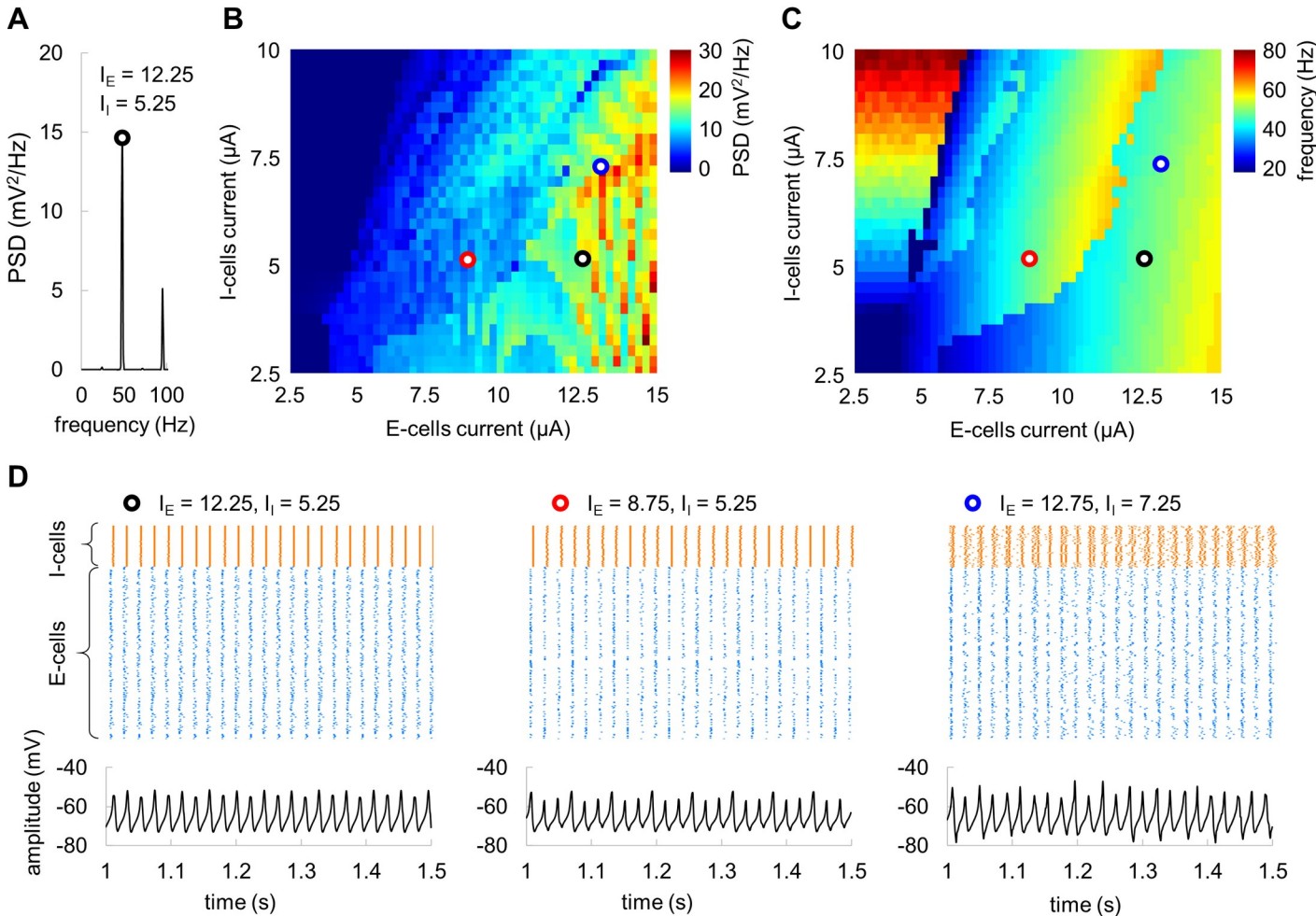

**Fig 6. Model parameter space.** (A) The PSD with spectral peak at 48 Hz for input currents $I_E$ = 12.25 μA and $I_I$ = 5.25 μA. (B, C) Power at the spectral peak (B) and corresponding frequency (C) of the network oscillations as a function of input current to E-cells and I-cells. Black, red, and blue circles indicate pairs of the currents producing oscillations at 48 Hz. (D) Spiking activity and corresponding average membrane potentials for three selected input currents that produce oscillations at 48 Hz.

broadband input current, the model produced neuronal activity (Fig 8A, 8B, 8C and 8D) similar to those of the constant input currents (see, Fig 5). In the presence of broadband input, spiking activity of E-cells remained highly synchronised (Fig 8B), so that average membrane potentials showed oscillations that can be readily observed in population response (Fig 8C) and as well as in the power-spectral density (Fig 8D). By computing the TRF between the input broadband current and output voltage, we observed a gamma echo (Fig 8E) similar to that in MEG data. Importantly, the frequency of the echo matched the resonance frequency of the model (Fig 8F). To obtain more robust results, the PSD and TRF were averaged over 20 trials.

We further assessed the impact of the amplitude of the broadband input current on characteristics of the gamma echo. The input current was modelled as a sum of constant current of 12.25 μA and random (uniform) noise with amplitude ranged from 2 to 9 μA to the E-cells and a constant current of 5.25 μA to I-cells. The gamma echo revealed by MEG was resembling the model gamma echo for broadband input current of 4 μA (Fig 9). In case of lower currents

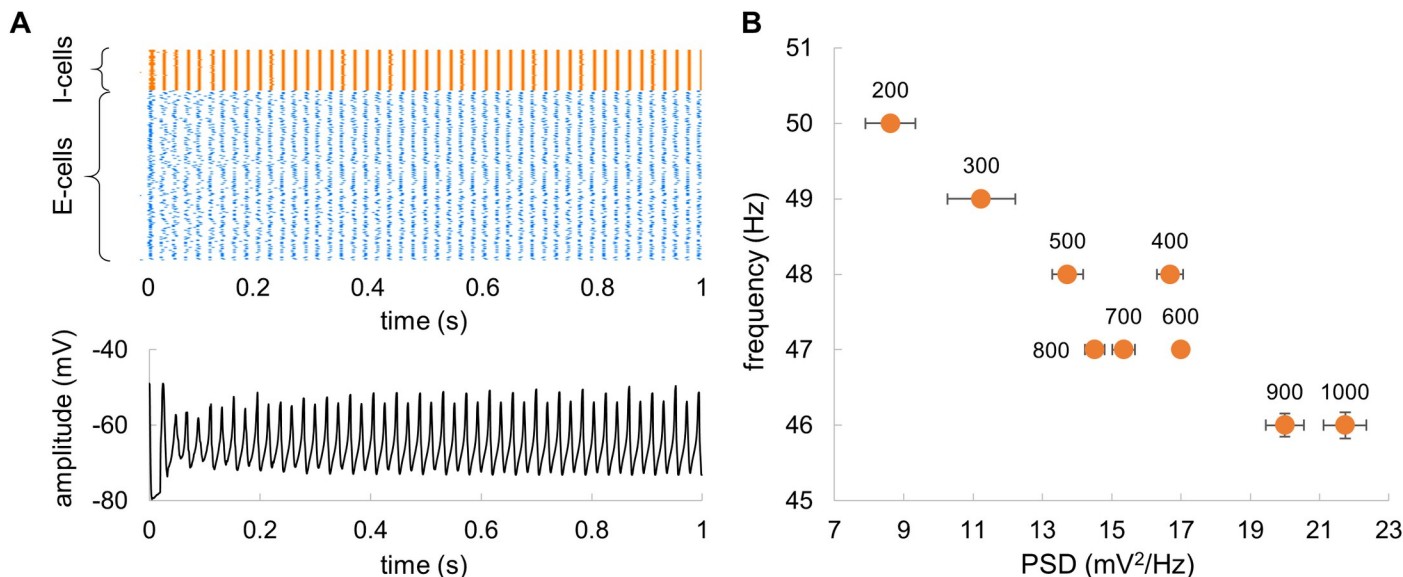

**Fig 7.** (A) Transients in spiking activity and average membrane potential after applying constant input currents to E-cells and I-cells. (B) The power (at the spectral peak) and corresponding frequency of the network as a function of the network size. The labels indicate the number of neurons in the network. Bars indicate standard deviation estimated in 20 trials.

(2 μA and 3 μA), the response at resonance frequency showed much longer decay. Conversely, the higher currents (5 μA to 9 μA) produced a response with shorter decay time.

## Oscillatory inputs produce maximum power at the resonance frequency

We further explored the model response by applying an oscillatory input current to E-cells in a similar manner as the broadband input current. The oscillatory input was modelled as a sum of constant current of 12.25 μA and a sine wave with amplitude of 9 μA to E-cells, while keeping the current to I-cells constant at 5.25 μA (Fig 10A). The oscillatory input increased synchronization among the E-cells in the gamma band (Fig 10B) compared to the absence of oscillatory input (see, Fig 5C). The average membrane potential of the E-cells also showed a larger amplitude (Fig 10C) compared to that of the constant input current (Fig 5D). The oscillatory input current (mimicking visual stimulation) at the resonance frequency (48 Hz) produced stronger response compared to a response at non-resonance (e.g. 78 Hz) frequency input (Fig 10D). To assess the spectral profile of the model in response to input currents of different frequencies, we applied a sinusoidal input current ranging from 1 to 100 Hz with 1 Hz steps and amplitude of 9 μA. The results showed an amplified peak in the power spectral density near the resonance frequency of 48 Hz (Fig 10E). Consistently with earlier findings [37], the spectral profile peaked at the resonance frequency and showed a decay towards higher frequencies.

To further investigate the model response to an oscillatory input, we computed the model output power (at the spectral peak) and corresponding frequency as a function of the input current, by systematically varying its frequency (1–100 Hz) and amplitude (0.5–10 μA). The results showed that the peak frequency in the spectral profile increased with the amplitude of input current (Fig 11A), meaning that the peak at around 48 Hz can be obtained for a specific input current of 9 μA (corresponds to Fig 10E). This suggests that the amplitude of visual input may differently affect frequency of the neuronal response. Interestingly, there was a minimum input current of 5 μA, above which, the frequency of the network oscillations matched

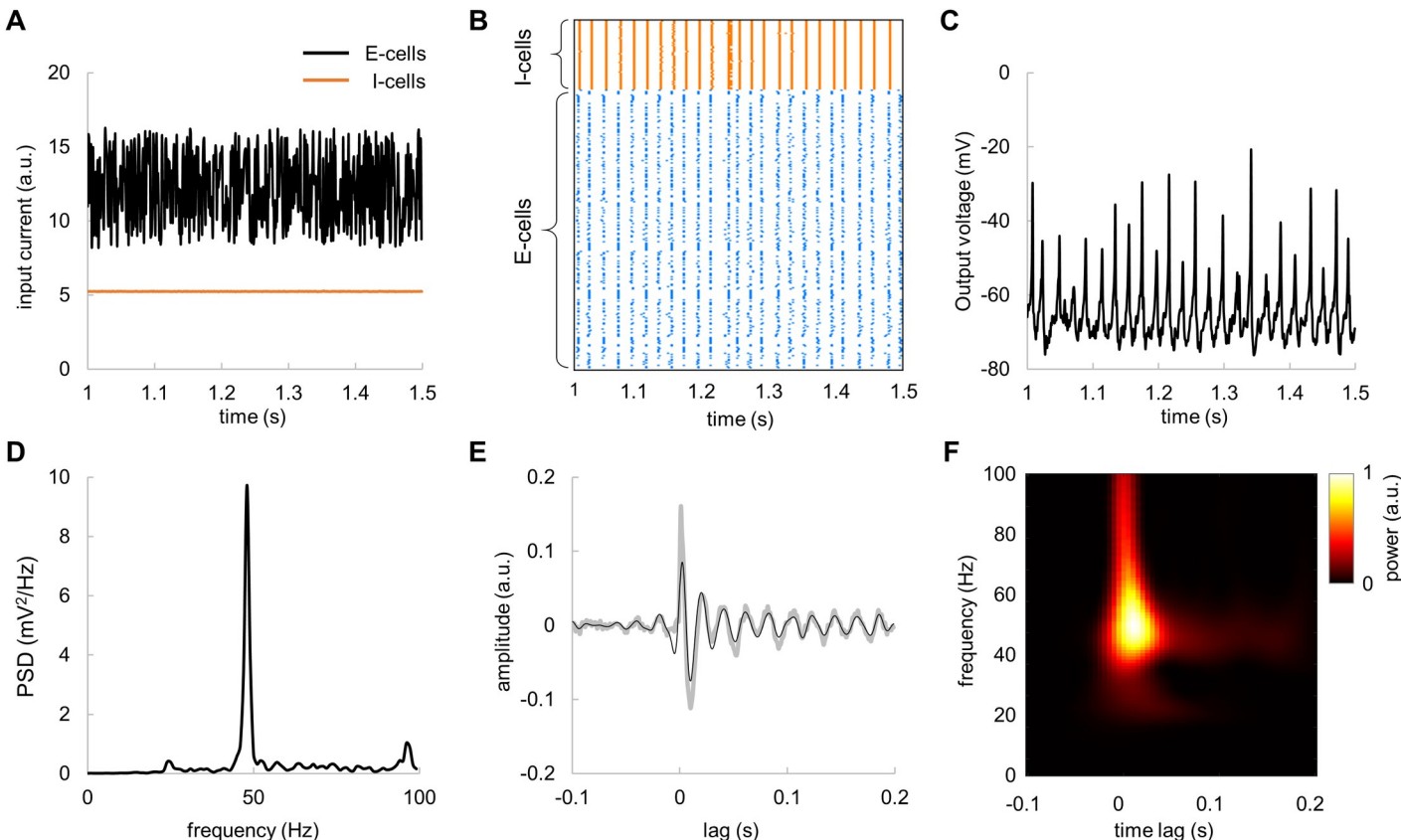

**Fig 8. Broadband input current to E-cells produces damped oscillations in the TRF–the gamma echo.** (A) Broadband input current to E-cells (black line) and constant input current to I-cells (orange line). (B) Spike rastergram for E-cells and I-cells for broadband input current. (C) The average membrane potential of the E-cells in response to fluctuating input currents. (D) Power spectral density of average membrane potentials of the E-cells averaged over 20 trials. (E) TRF assessed for the average membrane potentials of the E-cells with respect to the broadband input current. Note the clear "gamma echo". Gray and black lines depict respectively raw and the filtered TRF (40–100 Hz) averaged over 20 trials. (F) Time-frequency representation of power of the TRF.

the frequency of the input current (Fig 11B). This suggests that the frequency of external stimulation may not be directly translated to the firing rate.

## Discussion

In this study, we used broadband visual stimulation combined with MEG to assess the dynamical properties of the human visual cortex. We did this by estimating the temporal response function (TRF), i.e. the kernel best explaining the MEG signal from visual cortex when convolved to the broadband visual input. The TRF is similar to the cross-correlation function [38] in case of a random temporally uncorrelated input. In the TRF we observed an early response limited to the gamma band that we term the *gamma echo*. We also observed the known *perceptual echo* in the alpha band [9]. To explore the neuronal mechanisms producing the gamma echo, we implemented a biophysically plausible pyramidal-interneuron-gamma (PING) model. When driving the model by a broadband input current and estimating the respective TRF, we observed a gamma echo similar to that in the MEG data. Based on these simulations, we suggest that the gamma echo is produced by a network in which the dynamical properties are largely determined by GABAergic interneurons and their interaction with pyramidal cells; i.e. a PING-type network adjusted to produce damped oscillations in the gamma band can account for the gamma echo.

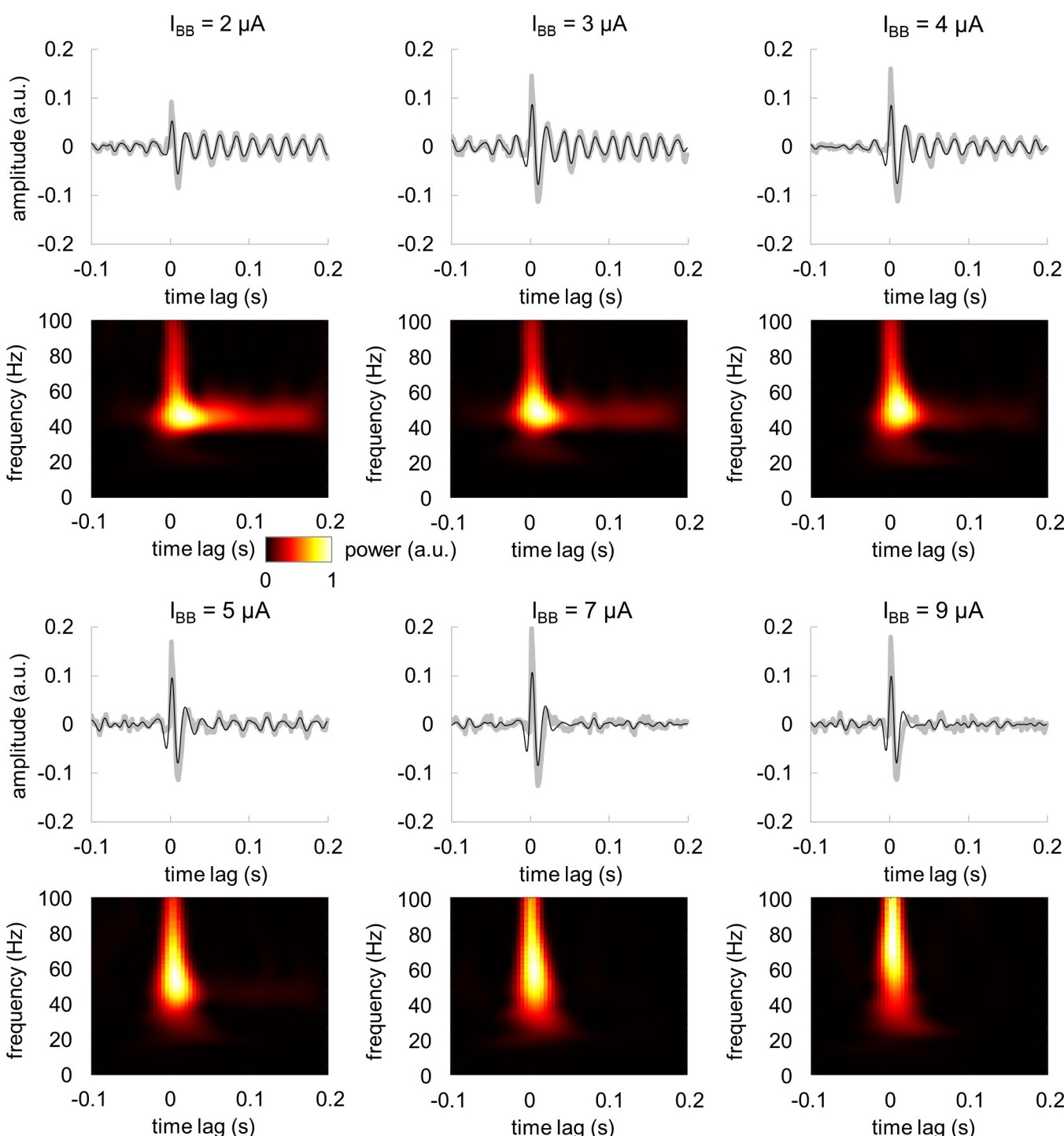

**Fig 9. The amplitude of the broadband input current determines temporal characteristics of the gamma echo.** TRF (*top panel*) and its time-frequency representation (*bottom panel*). $I_{BB}$ denotes amplitude of the broadband input current.

## Relationship between alpha and gamma echoes

Our findings reveal that both the alpha and gamma echo reflect the intrinsic properties of the visual system. However, there are several differences between the alpha and gamma band

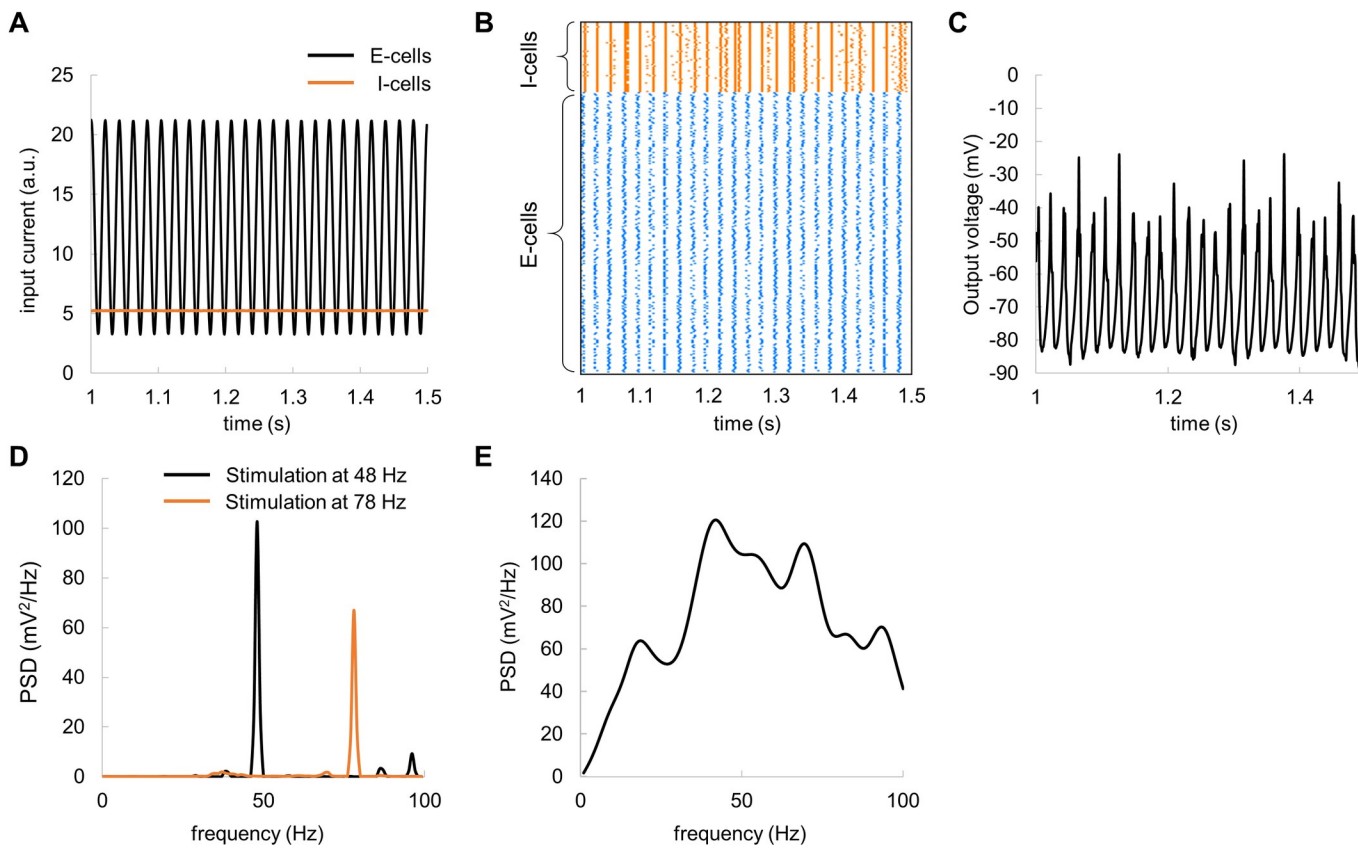

**Fig 10. Oscillatory input current reveals an amplification of oscillatory dynamics at the ~48 Hz resonance frequency.** (A) Oscillatory input current to E-cells (black line) and constant current to I-cells (orange line). (B) Spike rastergram for E-cells and I-cells. (C) The average membrane potential of the E-cells. (D) Example of power spectral density for input current at 48 Hz (resonance frequency, black line) and at 78 Hz (non-resonance, orange line). (E) Power spectral density of the average membrane potential for oscillatory input over multiple frequencies, 1–100 Hz, and amplitude of 9 μA.

echoes. First, while alpha echo occurs at around 0.2 s, gamma echo has a much earlier onset at 40 ms. Second, the gamma echo is largely localized in the primary visual cortex, whereas the alpha echo propagates over the cortex [39]. Third, the alpha-echo spans for up to 10 cycles, while gamma-echo vanishes after 2–3 cycles. These different characteristics suggest that the echoes are generated by different mechanisms and may not always occur in the brain concurrently.

At the same time the alpha and gamma echoes share similar properties. Earlier studies have demonstrated that the alpha echo shows maximum power at the individual alpha frequency [9]. Similarly, we found that the central frequencies of the gamma echo were specific to the participants. Our simulation results showed that the gamma echo of the model was strongly related to the resonance properties of the neuronal network. Thus, we suggest that the gamma echo reflects the intrinsic resonance properties of the early visual system.

Our results show that the gamma echo originates in the primary visual cortex and does not appear to propagate outside of this area. Conversely, a recent study showed that alpha echoes have the characteristics of travelling waves [39]. The authors further suggest that the spatio-temporal dynamics of the alpha echo is consistent with the timing of the neuronal activity that one might expect for feedforward and feedback communication associated with predictive coding. Indeed, both alpha and gamma oscillations might play a key role for predictive coding [40,41], and thus, propagation properties associated with the gamma echo may provide crucial insight on the feedforward dynamics associated with predictive coding.

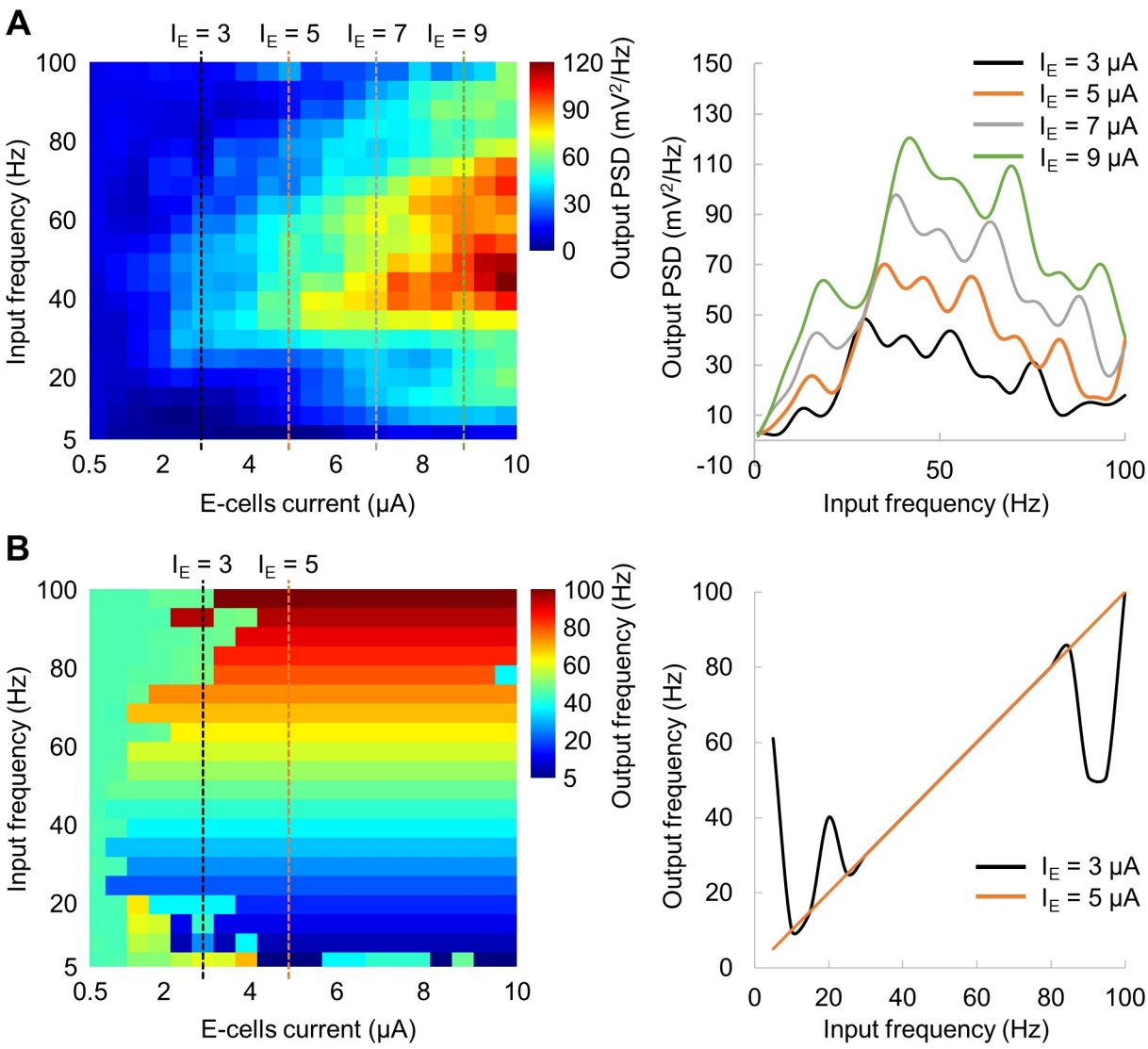

**Fig 11.** Network output power at the spectral peak (A) and corresponding frequency (B) as a function of the oscillatory input. The amplitude and frequency of the input current varied within 0.5–10 μA and 5–100 Hz, respectively. Note that the spectral profile on A (right panel) for $I_E$ = 9 μA corresponds to the spectral profile shown on Fig 10E.

In contrast to the alpha echo, the gamma echo has not been observed in previous studies that applied broadband visual stimulation (e.g. [9,42]), most likely since these studies relied on projectors with a relatively low refresh rate of 100 to 160 Hz. The resulting temporal resolution of 6–10 ms corresponds to at least 3 samples for a 48 Hz-cycle, and thus, it does not allow to fully capture gamma-band activity. In this study, we used a projector with a refresh rate of 1440 Hz which allows presenting stimuli with a sub-millisecond temporal resolution allowing for optimally estimating the gamma echo.

### Induced gamma oscillations and gamma echo

In this study, we presented grating stimuli concurrently with the broadband visual flicker, and hence, both the grating stimuli and the flicker can potentially induce a response in the gamma band. Analysing time-frequency representation of MEG power and the gamma echo, we

found that the visual grating induces gamma oscillations at around 57 Hz (grand average) whereas central frequency of the gamma echo was near 48 Hz, indicating that these phenomena do not overlap in the frequency domain. Furthermore, we did source localization of the induced gamma oscillations and the gamma echo. Although, the sources were located in the primary visual cortex closely to each other, there was an indication that sources of the induced gamma are slightly superior (~11 mm) than the sources of gamma echo. Consistently with our findings, previous studies [21,43–46] also demonstrated that static and dynamic gratings induce gamma oscillations in the primary visual cortex. Indeed, the gamma echo and induced gamma oscillations may originate from neighbouring but different areas (for instance, V1 and V2), which might not be easily dissociated due to limited spatial resolution of MEG. Based on the findings, we conclude that the gamma echo and the induced gamma oscillations are produced by different neuronal populations. This is consistent with the work of Duecker et al, 2020 [47] demonstrating the visual flicker does not entrain endogenous gamma oscillations but they are rather coexisting phenomena.

## Mechanism of the gamma echo

We implemented a biophysically realistic computational model based on the PING mechanism to clarify the underlying mechanism of the gamma echo. As we discussed earlier, the gamma echo and induced gamma oscillations are likely to be produced by different generators. This explains the absence of a resonant frequency in our recent study [47] which however was reported by Gulbinaite and colleagues [37]. In their study, the resonance frequency at about 47 Hz was observed when driving the visual system with rhythmic stimuli in a wide range of frequencies (3 to 80 Hz). We were able to reproduce this finding using our model and we therefore suggest that the rhythmic stimulation in study by Gulbinaite et al. did not affect the network producing endogenous gamma oscillations *per se*, but rather a network in V1 producing damped gamma oscillations. This way, the gamma echo does not reflect the dynamics of the endogenous gamma oscillations, but rather co-exists with endogenous gamma oscillations in different frequency bands.

## Computational model

In this study we used a relatively simple model to describe dynamics of the early visual cortex in response to the broadband and rhythmic stimulation. Our model was based on the Izhikevich framework [29] and thus, combined biological plausibility of Hodgkin–Huxley type dynamics and the computational efficiency of integrate-and-fire neurons. In addition, we improved the model by incorporating the kinetics of the AMPA and GABA neurons based on the formalism from Wang and Buszaki [11]. Since the gamma echo is well localised in the primary visual cortex (V1), and hence, the underlying mechanism does not require a complex interplay between areas in the visual system, our model adequately describe this phenomenon. In future work more complex multicompartment models [26] could also be utilised to describe gamma echo and perhaps provide some insight into the mechanism.

Our model was tuned to account for the gamma echo, and in future work it would interesting to extend the model framework such that the network can produce alpha oscillations as well. This would allow addressing more complex relationship between the gamma and alpha echoes from mechanistic perspective. Yet an avenue to explore would be to extend the models with a second network capable of producing endogenous gamma oscillations. This would allow for exploring the conditions when the network producing the gamma echo could also entrain that network producing the endogenous gamma oscillations.

### Implication of the findings

Our findings can provide new insight into the mechanism of evoked responses. It seems plausible that the characteristics of the early evoked responses in the visual system might be determined by the properties of the endogenous gamma activity. Indeed evoked gamma oscillations in response to visual stimuli have been identified in a previous study [48]. Importantly these oscillations increased with hypocapnia which is known to also increase GABAergic conductivity. In future work it would also be interesting to further uncover which components of visual evoked fields can be explained by the neuronal dynamics also producing the gamma echo [49].

### The gamma echo is not related to 50 Hz line noise

One might be concerned that the gamma echo is partly a consequence of the 50 Hz line noise given the similar frequencies. However, our analysis on magnetometers (see, S1 Fig) clearly demonstrates that suppression of the 50 Hz line noise does not change the characteristics of the gamma echo in any of the participants. Moreover, the gamma echo did vary from 46 to 56 Hz over participants. We conclude that the gamma echo is not biased by the 50 Hz line noise.

## Conclusion

Using broadband visual input stimuli we here provide evidence for a band-limited temporal response function in the gamma that we term the gamma echo. A computational model showed that a PING type of mechanism based on a network producing damped oscillations in the gamma band could account for the gamma echo. Nevertheless, the gamma echo is distinct from the mechanism producing endogenous gamma oscillations. The stage is now set for further investigating how the gamma echo is modulated by tasks such as spatial attention as well as uncovering how the echo might propagate in the visual hierarchy.

## Materials and methods

### Ethics statement

The study was approved by the local ethics committee (University of Birmingham, UK) and written informed consent was acquired before enrolment in the study.

### Participants

Five participants (mean age: 33; age range: 28–38; 1 female) with no history of neurological disorders partook in the study. All participants conformed to standard inclusion criteria for MEG experiments. Participants had normal or corrected-to-normal vision.

### Experimental paradigm

Two grating stimuli were presented bilaterally (Fig 12A). After 0.5 s from the stimuli onset, the left and right gratings started contracting for 3 s either coherently (same direction) or incoherently (different directions). The direction of the motion (up / down) of the left and right grating stimuli was random in consequent trials. The grating stimuli moved at a constant speed of 0.5 degree/s. The participants were instructed to focus on the fixation point and press the button when a cue indicating the direction of motion (i.e. coherent or incoherent) occurred at the fixation point.

The key novelty of the experiment is that the luminance of the left and right stimuli was modulated by two uncorrelated broadband (i.e., noise with uniform distribution) flickering

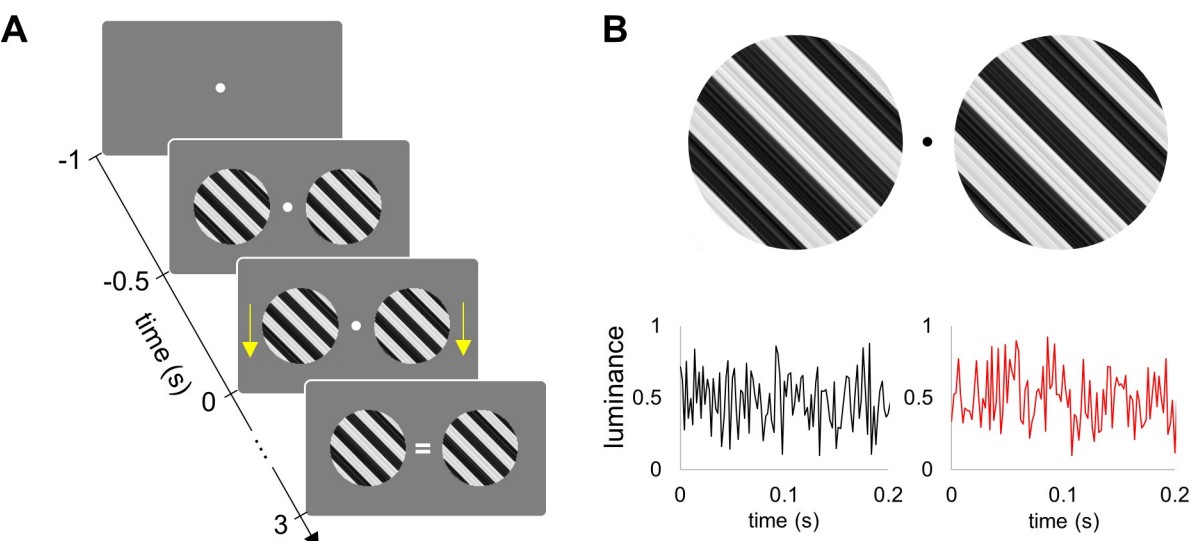

**Fig 12. Experimental paradigm.** (A) Grating stimuli were presented for 0.5 s and then the left and right gratings started contracting for 3 s either coherently (same direction) or incoherently (different directions). Yellows arrows on this figure (were not visible in the experiment) indicate coherent motion downwards. The cue (' = ') indicates coherent motion. (B) Luminance of the left and right grating stimuli was modulated by two independent broadband signals. The modulation (visual flicker) started at the time -0.5 s, together with the onset of grating stimuli.

signals (Fig 12B) at 1440 Hz. To this end, the grating stimuli were converted into textures using Psychophysics toolbox [50], and then the luminance of these textures was modulated by the flickering signals. We used the PROPixx DLP LED projector (VPixx Technologies Inc., Canada) to present the grayscale visual stimuli at a high refresh rate of 1440 Hz with a resolution of 960 x 600 pixels (see, [51]). Such refresh rate was achieved by presenting twelve frames within one refresh cycle of 120 Hz graphics card. The experimental paradigm was implemented in MATLAB 2018a (Mathworks Inc., Natick, USA), and the scripts are available on the OSF website (https://osf.io/fe8x5/).

## Magnetoencephalography data acquisition and processing

The MEG data were acquired using a 306-sensor TRIUX Elekta Neuromag system (Elekta, Finland). The magnetic signals were bandpass filtered from 0.1 and 330 Hz using embedded anti-aliasing filters and then sampled at 1000 Hz. The acquired time series were segmented into 4 s epochs; −1 to 3 s relative to the onset of the stimuli motion. Note that the stimuli were flickering during the -0.5 to 3 s time interval. Simultaneously with the MEG, we also acquired the eye-movements and blinks using an EyeLink eye-tracker (SR Research, Canada). Eye blinks were detected in the X-axis and Y-axis channels of the eye tracker by applying a threshold of 5 SD. The saccades were detected using a scatter diagram of X-axis and Y-axis time series of the eye-tracker for each trial. An event was classified as a saccade if the focus away from the fixation point by 2° and lasted longer than 500 ms. Trials contaminated by blinks and saccades were removed from further analysis. We also rejected trials containing large-amplitude events (above 5 SD) in MEG which are mainly associated with motion and muscle artefacts. As a result, the number of trials that remained after exclusion was 142 ± 10 (mean ± SD) per participant. For each participant, the number of trials per condition was equalized by randomly selecting the same number of trials. Such equalising serves to avoid a potential bias in TRF estimation related to unbalanced number of trials.

## Power spectral density

The power spectral density was estimated using Welch's method as implemented in the SciPy toolbox [52]. To this end, 4 s epochs of data were divided in 1 s segments with 50% overlap and weighted by a Hanning taper. The Fourier transform was applied to each segment and squared Fourier coefficients were averaged over the segments.

The same approach was applied to estimate the power spectral density of the modelled data.

## Temporal response functions

Temporal response functions (TRF) were estimated using ridge regression as implemented in mTRF toolbox [38]:

$$TRF = (S^T S + \lambda I)^{-1} S^T x$$

where $S$ is the lagged time series of the stimulus, $x$ denotes neuronal response, $I$ is the identity matrix, and $\lambda$ is the smoothing constant or "ridge parameter". In this study, the smoothing constant $\lambda$ was set to 1. Note that TRF is similar to a linear cross-correlation function if the stimulus is a random (temporally uncorrelated) signal.

The TRFs were computed between the broadband flickering signal and the MEG gradiometer with the strongest visual flicker response. Note that the TRF in Fig 1 was computed for an occipital gradiometer that captures signals in both the alpha and gamma bands. In order to assess the contribution of the 50 Hz line noise to the TRF, we also computed the TRF for MEG magnetometer with strongest response to the visual flicker before and after applying source space separation (SSS) method [34] for noise reduction. For the modelled data we calculated the TRF between the broadband input current and the average membrane potential for the E-cells.

## Time-frequency analysis

The time-frequency representations of power of the TRF were computed using the Hanning taper approach as implemented in the Fieldtrip toolbox [53]. We used time-windows of different length spanning 5 cycles at the specific frequency. The analysed frequency range was 5–100 Hz with steps of 1 Hz and the time ranged from -0.1 to 0.2 s (or 0.7 s as in Fig 12) with steps of 5 ms. In case of the induced gamma oscillations (see, Fig 3), we reported a relative change in MEG power during the -0.5 to 3 s stimulation interval compared to the -1.0 to -0.5 s baseline as follows, $P_{rel} = (P_{ST}—P_{BL}) / (P_{ST} + P_{BL})$, where $P_{ST}$ and $P_{BL}$ denote MEG power during stimulation and baseline, respectively.

## MRI data acquisition

A high-resolution T1-weighted anatomical image (TR/TE of 7.4/3.5 ms, a flip angle of 7˚, FOV of 256×256×176 mm, 176 sagittal slices, and a voxel size of $1{\times}1{\times}1$ mm$^3$) was acquired using 3-Tesla Phillips Achieva scanner.

## Source reconstruction

To build a forward model, we first manually aligned the MRI images to the head shape digitization points acquired with the Polhemus Fastrak. Then, the MRI images were segmented, and a single shell head model was prepared using spherical harmonics fitted to the brain surface [54]. The individual anatomy was warped into standard MNI template using the Fieldtrip toolbox [53].

To localise power of the of induced gamma rhythm in source space, we used the Dynamical Imaging of Coherent Sources (DICS; [55]) approach as implemented in the Fieldtrip toolbox. The time-frequency analysis was applied to the MEG data in the -0.5 to 3 s interval that covers the entire duration of the flickering stimuli.

To localise the gamma echo response, we used Linearly Constrained Minimum Variance (LCMV) beamformer [56] as implemented in the Fieldtrip toolbox. To this end, we first reconstructed time series in source space by applying LCMV beamformer to MEG data, and then estimated TRF (see, above) for each source point. The covariance matrix for the LCMV beamformer was estimated for bandpass filtered MEG data (40–100 Hz) in the -0.5 to 3 s time interval.

The location differences between sources of induced gamma and gamma echo response were assessed by extracting the coordinates (along the interior-superior z-axis) of these sources for each participant separately, and then applied the t-test over participants.

## Model

We modelled the neuronal populations of cortical areas as a network of interconnected excitatory and inhibitory neurons (Fig 5A). The network model was composed of 400 regular spiking excitatory pyramidal neurons (E-cells), and 100 fast-spiking inhibitory interneurons neurons (I-cells). The number of neurons as well as ratio between E-cells and I-cells (4 to 1) are consistent with previous studies [29,31].

## Neuronal model

We used the neuronal model proposed by Izhikevich [29] to simulate the membrane potentials of the excitatory and inhibitory neurons.

$$v' = 0.04v^2 + 5v + 140 - u + I + I_{syn} \tag{1}$$

$$u' = a(bv - u) \tag{2}$$

$$s'_{AMPA} = \alpha_{AMPA}F(v)(1 - s_{AMPA}) - \beta_{AMPA}s_{AMPA} \tag{3}$$

$$s'_{GABA} = \alpha_{GABA}F(v)(1 - s_{GABA}) - \beta_{GABA}s_{GABA} \tag{4}$$

where $v$ represents the membrane potential of the simulated neuron, $I$ determines the input current, $u$ is a slow recovery variable. The model also includes a reset: when $v$ exceeds 30 mV, an action potential is assumed, and the variables are reset: $v = c$ and u = u + d. The coefficients a = 0.02 and b = 0.2, c = -65 and d = 8 define the regular spiking E-cells, while a = 0.1 and b = 0.2 c = -65 and d = 2 define the fast spiking I-cells. The variable $s$ represents the gating for synaptic input and includes both $s_{AMPA}$ and $s_{GABA}$ defined for each sending E-cell and I-cell, respectively.

To model the kinetics of the AMPA and GABA neurons, we followed the formalism from Wang and Buszaki [11].

The term $I_{syn}$ reflects the synaptic current in the receiving neurons whereas $s$ reflects the gating variable in the sending neuron,

$$I_{syn} = \sum_{i=1}^{N}C_{(:,i)} \cdot s_{AMPA} \cdot (v_{AMPA} - v) + \sum_{j=1}^{M}C_{(:,j)} \cdot s_{GABA} \cdot (v_{GABA} - v) \tag{5}$$

where N is the number of excitatory neurons, M is the number of inhibitory neurons, C is the connectivity matrix, $v_{AMPA}$ and $v_{GABA}$ are reversal potentials of AMPA ($v_{AMPA}$ = 0 mV) and

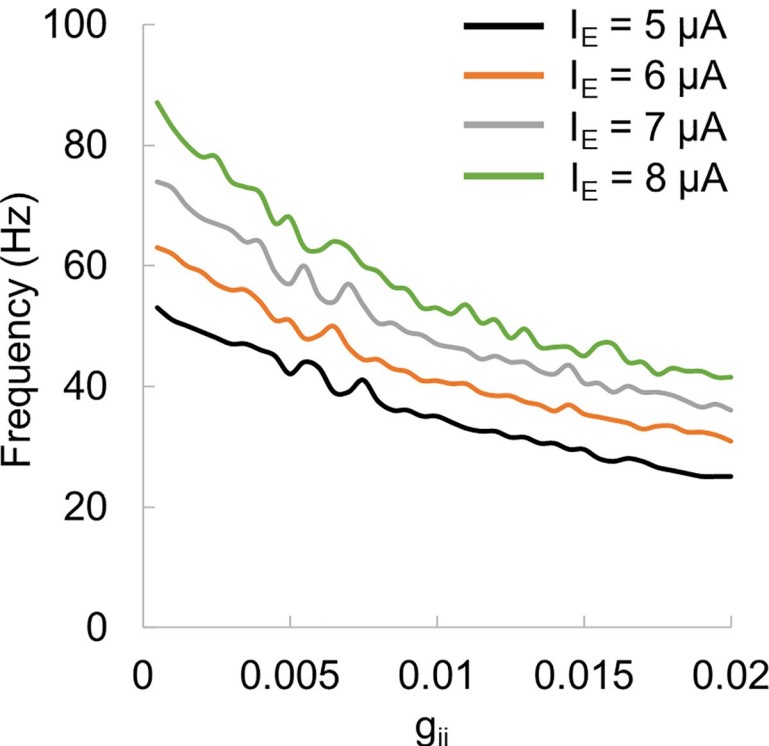

**Fig 13. Model output frequency as a function of connectivity strength between I-cells.**

GABA receptors ($v_{GABA}$ = -70 mV on E-cells and $v_{GABA}$ = -75 mV on I-cells), respectively. The differential Eqs (3 and 4) includes parameters channel opening rate $\alpha_{AMPA}$ = 12 (ms$^{-1}$) and the channel closing rate $\beta_{AMPA}$ = 0.5 (ms$^{-1}$) for AMPA receptors, and $\alpha_{GABA}$ = 12 (ms$^{-1}$) and $\beta_{GABA}$ = 0.1 (ms$^{-1}$) for GABA receptors; $F$ denotes a sigmoid function: $F(v) = 1/(1+exp(-v/2))$.

We specified the connectivity strength between different types of cells based on prior experimental results [36]. These connection strengths represented the amount of current that enters the receiving neuron after a spike of the sending neuron [31]. To make sure that the model connectivity was selected adequately, we assessed the model output frequency (firing rate) as a function of connectivity between I-cells for several input currents to E-cells and I-cells (Fig 13). The plot shows a decrease in frequency with increasing inhibitory connectivity strength as predicted by the PING mechanism [11].

We further adjusted the connectivity matrix while preserving the connectivity strength ratio between different type of cells, to obtain robust oscillations at 48 Hz for given input currents. The connectivity between cells was as follows: connectivity between I-cells ($c_{ii}$ = 0.004), connectivity from I-cells to E-cells ($c_{ie}$ = 0.006), and connectivity from E-cells to I-cells ($c_{ei}$ = 0.003). Since the connectivity strength between E-cells is much lower compared to other cells, ~0.25 * $c_{ii}$ [31], we set connectivity between E-cells to 0 ($c_{ee}$ = 0.0).

## Population activity and local field potential produced by the model

The population activity of the model reflecting the local field potentials was computed by averaging the membrane potentials of the E-cells. This somehow approximates the fields measured by MEG which are generated by the sum of dendritic currents in pyramidal cells [57].

To solve the differential Eqs (1–4) numerically, we used the Euler method with the timestep $\Delta t$ = 1 ms.

## Supporting information

**S1 Fig. Suppression of line noise in data does not change the characteristics of the gamma echo in representative magnetometers.** Each panel shows the power spectral density (PSD) and TRF for individual participants before (blue line) and after (red line) applying the SSS method to suppress 50 Hz line noise. The echoes remain strong after the 50 Hz line noise is suppressed.
(TIF)

## Author Contributions

**Conceptualization:** Alexander Zhigalov, Katharina Duecker, Ole Jensen.

**Data curation:** Alexander Zhigalov, Katharina Duecker.

**Formal analysis:** Alexander Zhigalov.

**Funding acquisition:** Ole Jensen.

**Investigation:** Alexander Zhigalov.

**Methodology:** Alexander Zhigalov, Ole Jensen.

**Project administration:** Ole Jensen.

**Resources:** Ole Jensen.

**Software:** Alexander Zhigalov, Ole Jensen.

**Supervision:** Ole Jensen.

**Validation:** Alexander Zhigalov.

**Visualization:** Alexander Zhigalov.

**Writing – original draft:** Alexander Zhigalov, Ole Jensen.

**Writing – review & editing:** Alexander Zhigalov, Katharina Duecker, Ole Jensen.

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
