## [Decision Letter · Decision Letter 0]

4 Feb 2021

Dear Dr. Zhigalov,

Thank you very much for submitting your manuscript "The visual cortex produces a signal in the gamma band in response to broadband visual flicker" for consideration at PLOS Computational Biology.

As with all papers reviewed by the journal, your manuscript was reviewed by members of the editorial board and by several independent reviewers. The paper was overall very well received, still some points need clarification. In light of the reviews (below this email), we would like to invite the resubmission of a revised version that takes into account the reviewers' comments.

We cannot make any decision about publication until we have seen the revised manuscript and your response to the reviewers' comments. Your revised manuscript is also likely to be sent to reviewers for further evaluation.

Sincerely,

Daniele Marinazzo

Deputy Editor

PLOS Computational Biology

Reviewer's Responses to Questions

**Comments to the Authors:**

Reviewer #1: Summary :

The authors report a novel oscillatory response function in the gamma band (~48Hz) by stimulating the visual cortex with white noise visual flicker. They also implement a PING model that replicates the same dynamics. Overall the paper is clearly written, and the findings are of interest for neuroscientists investigating oscillatory dynamics. I only have a few minor points that hopefully could help improve the manuscript.

Main points :

- one main concern about the presentation of these results is the lack of representation of the gamma echoes topography. Besides specifying precisely the location of each sensor used in the analysis for each subject, it would be very interesting to show these echoes' cortical extent. Are they localized mostly in occipital regions, or do they also appear in more central/frontal ones? Besides, the PING network does not necessarily model visual areas, and it could be interesting to generalize your results to other sensory (or non-sensory) cortical regions.

- It is not clear to me the difference between endogenous gamma and gamma echoes (and flicker responses). Related to the previous point, would it be possible to localize precisely to V1 the gamma echoes, to support the authors' claim that these are originated in V1, whereas endogenous gamma is originated in V2? Also, what would be the functional role –if any- of gamma echoes and endogenous gamma oscillations? In this light, why would the endogenous gamma be higher in the visual hierarchy (as suggested by the authors)?

- Your simulation shows that an oscillatory input at the right resonance frequency (48Hz) is amplified by the network, producing an amplified peak in the power spectral density. Isn't this result in contradiction with previous findings showing that visual stimulation may not entrain gamma oscillations? (reference [46] in the paper, from the same senior author as the current study).

- The model's coefficients and parameters have not been explored at all in this study. However, it could be interesting to investigate how the TFR changes properties as a function of some relevant parameters. Ideally, this could shed some light on the differences between participants (peak frequency, amplitude, etc.). Additionally, it would be best to motivate the choice of the parameters' value (are they all the same as in the cited Izhikevich model ?) and their biological plausibility. Would the coefficients' value be different in other brain regions (e.g., higher in the visual system), thus predicting differences in the TFR?

- Why equalizing the number of trials per condition in each participant? Isn't it reasonable to assume that the two conditions (coherent and incoherent motion) are orthogonal to the TRF?

- The Results section's first paragraph mentions that figure 2B is from an occipital sensor of a representative participant. However, none of the subplots in figure 3 (with all participants) looks like the same image, despite having the same scale.

- It would be interesting, in the next study, to replicate these results on a larger population. In alpha-band echoes, some participants don't manifest any reverberation, possibly due to the cortex's anatomical configurations. If this is not the case in the gamma echoes (from figure 3, it seems that 5 participants out of 5 show them quite clearly), this could provide some indications that different mechanisms are involved in the two frequency bands.

- Related to the previous point, it could be interesting to discuss/speculate about the differences between the echoes in the two frequency bands: do they both reflect two resonance properties of different neuronal networks, the gamma one more locally and the alpha one more globally? Or do they underlying different mechanisms and functions?

Minor details :

- in the 'power spectral density' the epochs are defined as 5 seconds long; in the previous paragraph, it was 4. Which one is it?

- It seems that some figures are referenced in the wrong place. The 'Model' paragraph refers to figure 2A, but the model is in figure 4. Similarly, later the connectivity matrix refers to figure 3A, but I see it in 4B.

Reviewer #2: In this manuscript the authors applied a broadband flicker (1-720 Hz) to drive human visual cortex while measuring the MEG response. Moreover they estimated the temporal response function (TRF) between the visual input and the MEG response, which revealed an early response in the 40-60 Hz gamma range as well as in the 8-12 Hz alpha band.

Simulations of a network model for gamma oscillations with a broadband signal complete the results, thus allowing the authors to estimate numerically the TRF and to compare it with the one from MEG study.

The writing of the article is often sloppy and many paragraphs need to be re-written in order to improve clarity. Some examples are reported in the detailed list of comments/questions in the following. Moreover I find the Introduction and the Discussion too focused on the experimental results, while a sufficient literature regarding the state-of-art of the models is neglected. All in all the results of this manuscript are more oriented towards numerical experiments than the experimental ones, thus creating an imbalance in the discussion.

In the following is reported a list of comments/questions.

INTRODUCTION

1. Citing the text:

Furthermore, the GABAergic feedback also serves to synchronize the population activity [14,15] "The pyramidal neurons

also play an important role in the mechanism." In each cycle [...]

The sentence reported in "..." seems to have been forgotten there.

2. [..] we recorded the MEG while stimulation the visual system using a broadband (1-720 Hz) visual flicker [..]

Please replace "while stimulation" with "while stimulating"

MODEL

3. Why do you use such a limited number of neurons (200 in total)?

4. How is chosen the connectivity matrix?

RESULTS

5. "The temporal response function (TRF) of a system..."

The acronym TRF has been already introduced before; there is no necessity to re-introduce it now.

6. Merge the paragraphs "TRF show individual resonance frequency of gamma echo" and "The gamma echo is close to 50 Hz but not due to line noise" for a better comprehension of the results shown in Fig. 3. At the moment it is necessary to read the latter paragraph in order to understand what is written in the previous one, where Fig. 3 is cited.

7. Could you better explain the motivations underlying the choise of connection strength?

In the text is written:

"we used a relatively low connection strength between E-cells (cee = 0.001), high connection strength between I-cells (cii = 0.200), while keeping connectivity between E-to-I-cells (cei = 0.050) and I-to-E-cells (cie = 0.010) at a moderate level"

A biophysical justification would help.

8. Fig. 4: Merge panels C and D. The information now contained in panel C is negligible. Add in panel A the symbols for 2 external curves impinging both E and I-cells.

9. Fig. 4: If you simulate for longer times, do you still see oscillations? Locking at panel E, it seems that the amplitude of the oscillations is shrinking, thus suggesting a possible transient phenomenon.

10. Fig. 4: A dependence of the results on the amplitude of the external current need to be shown in order to justify the choise of 25 pA.

11. Fig. 5: How much do the results depend on the amplitude of the noise on the E-cells?

What happens for smaller and bigger amplitude values with respect to 16pA?

12. The definition of the external current as "a sinusoidal function with a given frequency (0 - 100 Hz)" is misleading. It should be clearly mentioned the employed value (or values) of the used frequency.

13. Fig. 6: In panel C it is calculated the average membrane potential of the E-cells. Is it the same also for the previous figures? If yes, please write it explicitly.

14. Fig 6: Which is the range of observability of the shown phenomenon? If you vary the frequency of the external current, in which frequency range do you still observe it?

Are you able to observe both alpha and gamma peaks at the same time?

From what I see observing panel D, the emergent frequency is simply the frequency of the external stimulation.

15. Fig. 6: is the process shown in panel C stationary? What does it happen if you simulate for longer times?

DISCUSSION

16. Please replace "Using broadband visual input stimuli were here provide evidence for [..}"

with "Using broadband visual input stimuli we here provide evidence for [..]"

To conclude, the manuscript cannot be published in this form: the text is sloppy and a detailed investigation of the phenomenon is not presented. In order to understand the origin of the phenomenon, a deeper investigation has to be provided. At the moment I just see (probably) transient phenomena, that are shown just for few exemplary cases, without giving a clear view on the dependence on the amplitude and the frequency of the external stimulation.

Reviewer #3: The authors used a novel visual gratings task where subjects were asked to assess motion congruence/incongruence per trial. The left and right visual stimuli (presented to the left and right of fixation) were generated such that luminance was varied using orthogonal broadband random signals (i.e. noise with uniform distribution) flickering signals. MEG was recorded in 5 adult subjects. The Temporal response function (TRF) showed a subject specific gamma echo (40-60Hz) as well as an alpha (8-12Hz) echo. The dominant frequency of the novel gamma echo was subject specific and thus likely a reflection of early visual cortex activity. A PING model was designed to then mimic these TRF responses, lending support for contribution of GABA signaling in the early processing of visually presented stimuli.

Impression: The experiment and modeling findings are interesting. I did think that the methods description regarding the visual stimuli could have included more detail. In addition, the implications of these findings to prior knowledge regarding visual processing could be discussed further (implications wrt perception/ visual acuity/ evoked responses, etc. or new opportunities to evaluate visual impairment, etc. might be discussed). As written, we just get the sense that emitted and echoed gamma responses arise from different visual processing areas. How can we use this new information be used to study/explain visual processing?

Q1: Why is it called the “alpha band perceptual echo”? Is conscious perception necessary?

Q2. Relatedly, why are you using the term “gamma echo” and not the “gamma perceptual echo”?

Q3. In Figure 1, I can see the luminance values of each gratings stimulus are different [uncorrelated] between left and right grating stimulus; were the stimuli also uncorrelated [random?] on each trial as well?

Q4. I don’t understand how these luminance signals were used to produce the flicker stimuli. Can you give a more detailed description of the methods rather than simply referring to reference [26]?

Q5. So, what does the brain response look like at the sensor level? Does the brain response usually appear as a 2 dipole field waxing in amplitude in relation to the luminance of the independent stimuli? It reads that there was a single max gradiometer selected per subject. Shouldn’t there be one per hemifield? Was the max observed in the TRF based on all trials (i.e., like an FFT?).

Q6. How did the selected sensor compare over subjects? Same hemisphere? Same Channel?

Q7. Emitted visual gamma oscillations from static gratings are typically localized to pericalcarine cortex (i.e., primary visual cortex; see Gaetz et al., 2011), which is at odds with the statement in the Conclusion: “…that the gamma echo is produced by early visual cortex (most likely V1) whereas the

endogenous gamma oscillations are produced slightly down-stream (e.g. V2)”.

**Have all data underlying the figures and results presented in the manuscript been provided?**

Reviewer #1: **No: **From the submission : "Data cannot be shared publicly because of confidentiality policy. Data are available

from the Centre for Human Brain Health, University of Birmingham, Institutional Data

Access / Ethics Committee (contact via a.zhigalov@bham.ac.uk) for researchers who

meet the criteria for access to confidential data."

Reviewer #2: **No: **Not all parameter values are given in the captions

Reviewer #3: Yes

PLOS authors have the option to publish the peer review history of their article (what does this mean?). If published, this will include your full peer review and any attached files.

Reviewer #1: **Yes: **Andrea Alamia

Reviewer #2: No

Reviewer #3: No
---

## [Decision Letter · Decision Letter 1]

6 May 2021

Dear Dr. Zhigalov,

We are pleased to inform you that your manuscript 'The visual cortex produces gamma band echo in response to broadband visual flicker' has been provisionally accepted for publication in PLOS Computational Biology.

Best regards,

Daniele Marinazzo

Deputy Editor

PLOS Computational Biology

Daniele Marinazzo

Deputy Editor

PLOS Computational Biology

Reviewer's Responses to Questions

**Comments to the Authors: **

Reviewer #1: The authors fully addressed all my previous questions.

Reviewer #2: The authors have improved the manuscript and answered my questions.

Reviewer #3: I think the Authors have done a nice job with revisions. To clarify my earlier comment regarding Visual Gamma localization to Cuneus in VI - I referred to Gaetz et al., 2011:

Functional and structural correlates of the aging brain: Relating visual cortex (V1) gamma band responses to age‐related structural change

William Gaetz Timothy P.L. Roberts Krish D. Singh Suresh D. Muthukumaraswamy

First published: 18 July 2011 https://doi.org/10.1002/hbm.21339Citations: 38

I leave it to the Authors/Editor to incorporate/clarify how the current findings support / contrast with these published findings (if they feel comment / inclusion is warrented).

**Have the authors made all data and (if applicable) computational code underlying the findings in their manuscript fully available?**

Reviewer #1: Yes

Reviewer #2: **No: **Numerical codes and experimental data are not available

Reviewer #3: Yes

PLOS authors have the option to publish the peer review history of their article (what does this mean?). If published, this will include your full peer review and any attached files.

Reviewer #1: **Yes: **Andrea Alamia

Reviewer #2: No

Reviewer #3: No

---

## [Editor Report · Acceptance letter]

27 May 2021

PCOMPBIOL-D-21-00044R1 

The visual cortex produces gamma band echo in response to broadband visual flicker

Dear Dr Zhigalov,

I am pleased to inform you that your manuscript has been formally accepted for publication in PLOS Computational Biology. Your manuscript is now with our production department and you will be notified of the publication date in due course.

With kind regards,

Zsofi Zombor
